# Randomized Longitudinal Study Comparing Three Nasal Respiratory Support Modes to Prevent Intermittent Hypoxia in Very Preterm Infants

**DOI:** 10.3390/children7100168

**Published:** 2020-10-05

**Authors:** Maximilian Gross, Anette Poets, Renate Steinfeldt, Michael S. Urschitz, Katrin Böckmann, Bianca Haase, Christian F. Poets

**Affiliations:** 1Department of Neonatology, University Children’s Hospital Tübingen, 72076 Tübingen, Germany; maximilian.gross@med.uni-tuebingen.de (M.G.); anette.poets@med.uni-tuebingen.de (A.P.); katrin.boeckmann@med.uni-tuebingen.de (K.B.); bianca.haase@med.uni-tuebingen.de (B.H.); 2Department of Pediatrics, Klinikum Neukölln, 12351 Berlin, Germany; renate.steinfeldt@arcor.de; 3Division of Paediatric Epidemiology, Institute of Medical Biostatistics, Epidemiology, and Informatics, University Medical Centre of the Johannes Gutenberg University Mainz, 55131 Mainz, Germany; urschitz@uni-mainz.de

**Keywords:** very low birthweight infant, nasal respiratory support, s-NIPPV, NCPAP, intermittent hypoxia

## Abstract

Nasal continuous positive airway pressure (NCPAP) devices using variable (vf-) and continuous (cf-) flow or synchronized nasal intermittent positive pressure ventilation (s-NIPPV) are used to prevent or treat intermittent hypoxia (IH) in preterm infants. Results concerning which is most effective vary. We aimed to investigate the effect of s-NIPPV and vf-NCPAP compared to cf-NCPAP on the rate of IH episodes. Preterm infants with a gestational age of 24.9–29.7 weeks presenting with IH while being treated with cf-NCPAP were monitored for eight hours, then randomized to eight hours of treatment with vf-NCPAP or s-NIPPV. Data from 16 infants were analyzed. Due to an unexpectedly low sample size, the results were only reported descriptively. No relevant changes in the rate of IH events were detected between cf- vs. vf-NCPAP or between cf-NCPAP vs. s-NIPPV. Although limited by its small sample size, s-NIPPV, vf- and cf-NCPAP seemed to be similarly effective in the treatment of IH in these infants.

## 1. Introduction

Nasal continuous positive airway pressure (NCPAP) is effective for treating apnea of prematurity (AOP) and preventing intermittent hypoxia (IH) [1]. It helps to keep the airways open [2] and improves oxygenation [3] as well as lung function [4]. NCPAP may also reduce the work of breathing, at least if applied via devices with variable gas flow (variable flow = vf-NCPAP) [5,6]. In addition to NCPAP, nasal intermittent positive pressure ventilation (NIPPV) is used to treat AOP or IH. In a meta-analysis, the latter was found to be more effective than NCPAP in preventing re-intubation, but only if synchronized with the infant’s own breathing efforts (s-NIPPV) [7,8,9].

In an earlier study, we compared continuous flow (cf-) with vf-NCPAP and NIPPV concerning the rate of hypoxemic episodes and observed less such episodes with vf-NCPAP. NIPPV in that study, however, was not synchronized to the infants’ own breathing efforts [10]. In the present study, we aimed to investigate the effect of s-NIPPV on the rate of IH episodes comparing cf-NCPAP with vf-NCPAP and s-NIPPV.

## 2. Materials and Methods

### 2.1. Patients

Between October 2014 and September 2019, infants admitted to the neonatal intensive care unit at Tuebingen University Children’s Hospital site were evaluated for their eligibility to participate in this study. Inclusion criteria were (i) gestational age (GA) at birth ≤34 weeks (w), (ii) postmenstrual age at study ≤38 w, (iii) persistent AOP, and (iv) frequent apneas, expressed as an apnea score ≥5 on a rating scale used in the unit [11], despite treatment with hospital pharmacy-produced caffeine base and cf-NCPAP. The apnea score assigned points depending on the response of the nursing staff to events involving bradycardia (heart rate < 80/min) or hypoxemia (SpO_2_ < 80%). If no intervention was needed, one point was given; two points if tactile stimulation was applied or a manual inspiration via the ventilator, and three points if the baby had to be turned over in preparation for bagging; eight points were given if the baby was felt to briefly need positive pressure ventilation to recover. The respective points were added up. Exclusion criteria were severe congenital malformations, neuromuscular conditions or chromosomal abnormalities, the presence of symptomatic apnea (e.g., due to sepsis, hypoglycemia or cerebral hemorrhage), or the lack of written informed parental consent.

### 2.2. Study Design

The study design included two different before–after studies involving one control and two test interventions (Figure 1). After having entered the study, infants continued on their standard cf-NCPAP device (Sophie, Fritz Stephan, Gackenbach, Germany; control intervention) to acquire at least eight hours of recording time. Only if they had reached a value of ≥10 on the above apnea score, they were randomized using sequentially numbered opaque concealed envelopes prepared by an assistant not involved in patient care to either vf-NCPAP (test intervention 1, Infant Flow Plus (Vyaire Medical Inc., Chicago, IL, USA) or s-NIPPV via a standard neonatal ventilator (test intervention 2, Sophie, Fritz Stephan). Starting with this initial eight hour run-in phase, infants had their heart rate and arterial oxygen saturation (measured by pulse oximetry; SpO_2_) recorded on a standard infant monitor (Vitaguard VG 3100, GeTeMed, Teltow, Germany) in 3–4 s averaging mode; these recordings continued without delay after the infants had been switched to their respective study ventilator. Otherwise, the infants received their routine care, including treatment with a fixed dose of caffeine base (5 mg/kg/day). No infants received doxapram.

### 2.3. Recordings of Physiological Signals

The above monitor stored the pulse rate and SpO_2_, as well as the pulse waveforms, perfusion index and signal quality (Signal IQ^®^: Signal Identification and Quality, Masimo, Irvine, CA, USA) continuously. These recordings were analyzed using proprietary software (VitaWin^®^, version 3.3, GeTeMed, Teltow, Germany) after completing the study recruitment, thus not influencing any clinical decisions. Recordings were evaluated manually and periods with an artifactual signal, defined as a signal IQ <0.2, excluded [12]. IH was defined as a decrease in SpO_2_ to <80% for >1 sec, bradycardia as a fall in heart rate to <80 beats per minute for more than one beat. The fraction of inspired oxygen (FiO2) and transcutaneous partial pressure of carbon dioxide (tcpCO_2_) were recorded via the unit’s patient data management system (PDMS; IntelliSpace, Philips, Eindhoven, Netherlands) that also included electronic documentation of the above apnea score as entered by the nursing staff. Average FiO_2_ and tcpCO_2_ were calculated as the mean of the values documented every 30 min by the PDMS.

### 2.4. Ventilators

While a standard neonatal ventilator (Sophie) was used to generate cf-NCPAP or s-NIPPV; vf-NCPAP was delivered using the Infant Flow Plus. The NCPAP level was set to 5 cmH_2_O throughout. S-NIPPV was delivered at a rate of 20 breaths/min and a peak pressure of 15 cmH_2_O. Synchronization with the infants’ breathing efforts was achieved by an abdominal pneumatic trigger capsule (Respiration Sensor, Fritz Stephan, Gackenbach, Germany). Respiratory support was delivered via binasal prongs of appropriate size (EasyFlow, Fritz Stephan, Gackenbach, Germany, or Infant Flow LP, Vyaire Medical Inc., Chicago, IL, USA).

### 2.5. Statistical Analysis

The primary endpoint was the rate of IH events per hour. Secondary endpoints were the rate of bradycardia events per hour, mean FiO_2_ and mean tcpCO_2_. Based on our previous study [10], we estimated that 26 patients per before–after study were required to detect a paired group difference of 30%, i.e., a reduction by 0.85 IH events per hour. However, patient recruitment took much longer than expected, so that the study ultimately had to be terminated after 27 infants had been recruited over five years (see below). Thus, the results were reported descriptively stratified by before–after study (median, minimum, maximum) and the treatment effects were estimated based on the mean of the individual differences (paired samples) and its corresponding 95% confidence interval (95%CI). No statistical hypothesis testing and no comparisons between vf-NCPAP and s-NIPPV were performed on this smaller-than-expected sample size.

### 2.6. Ethics Approval and Consent to Participate

The study was approved by the ethics committee of the medical faculty at the study site (No. 433/2014BO1). Written parental consent was obtained upon patient recruitment. This study was registered with the German Clinical Trials register DRKS00005387.

### 2.7. Supplementary Materials

The study protocol, CONSORT checklist and CONSORT flow chart are available online as Appendix A.

## 3. Results

Patient flow is shown in Figure 2, and patient demographics in Table 1. Regarding our primary endpoint, no clinically relevant changes in the rate of IH events were detected between cf- vs. vf-NCPAP or between cf-NCPAP vs. s-NIPPV (Table 2). The treatment effects (95%CI) were estimated to be 0.25 events per hour (−0.23–0.73; *n* = 10) for vf-NCPAP and −0.33 events per hour (−1.07 to 0.40; *n* = 6) for s-NIPPV. Treatment failure on the allocated device did not occur, i.e., no infant had to be intubated during the study period for apnea or bradycardia.

Comparing both continuous positive airway pressure (CPAP) modes, tcpCO_2_ was 3–4 mmHg lower during the vf-NCPAP or s-NIPPV support compared to cf-NCPAP, while FiO_2_ remained unchanged (Table 2).

## 4. Discussion

In this comparison of the effectiveness of vf-NCPAP and s-NIPPV in reducing IH rates in very preterm infants with AOP compared to standard treatment with cf-NCPAP, we found only a little difference between these three nasal respiratory support systems.

We had planned this study for a total recruitment of 52 infants. This proved impossible for several reasons: there were competing large interventional multicenter studies ongoing on the unit with recruitment taking place in the first one to two postnatal days, reducing the number of infants eligible for the present study. In addition, many infants who had initially reached an apnea score of ≥5 subsequently did not reach the score of 10 required for randomization to another nasal respiratory support system, so that they never qualified for randomization. The slow patient recruitment in part was also due to a lack of external funding, so that no dedicated research personnel was available to supervise the study. Thus, after extending the original recruitment period by more than two years, the study was eventually terminated prematurely, because recruitment would have lasted another eight years had it continued at the pace observed. This decision prohibited applying any statistical testing; nonetheless, the results showed only minor differences at a descriptive level, suggesting that either intervention had similar effectiveness.

Further limitations included the fact that we did not record apneas. However, in line with previous work from our group, we do not see this as a relevant limitation, as it is not the apnea, but its consequences, i.e., IH and to a lesser extent, bradycardia, that are relevant to the long-term outcome of preterm infants [13]. Regarding the effectiveness of s-NIPPV, synchronization between infant and ventilator depended on the nurses attaching the trigger capsule correctly. This was not systematically checked, but nurses on the unit were very experienced in using these capsules ensuring optimal placement and minimal trigger delay. Additionally, we used only a set level of pressure 5 cmH_2_O during the CPAP application. This reflected the unit policy, but may not have been sufficient to open the airway during obstructive apneas. Finally, infants’ postmenstrual age at the study was already 30.5 w, i.e., their apnea rate may already have decreased spontaneously.

Perhaps due to these limitations, the differences seen between the various nasal ventilator support systems were smaller than in other studies. For example, Gizzi et al., using a pneumotachograph-based system to synchronize the ventilator with the babies’ breathing efforts and focusing on both bradycardia and IH, found a 50% reduction in the rate of IH episodes during s-NIPPV compared to cf-NCPAP (median, 2.9 vs. 5.9/h, respectively) [9]. In our earlier study comparing vf- with cf-NCPAP, the differences in bradycardia–/IH-rates were also larger (2.8 vs. 5.4/h [10]). Thus, although the gestational age at birth and postmenstrual age at study were similar in all three studies, cardiorespiratory event rates were lower in the present study, but s-NIPPV again seemed to be more effective than vf-NCPAP. As averaging times on the pulse oximeters used were identical and apneas are still a common occurrence at the mean postmenstrual age of our study population [14], we have no explanation for the lower overall event rates other than this being a chance finding related to our small sample size. Moreover, the comparatively low IH rate and an overall low average oxygen demand may indicate a higher clinical stability in participating infants, which also may have hampered the detection of notable differences between interventions.

A different approach to nasal respiratory support for preterm infants has recently been introduced by the nasal application of the neurally adjusted ventilatory assist (NAVA) technique in preterm infants. Using this technique in eight preterm infants studied at a median post-menstrual age of 29 w, a 40% reduction in the number of episodes with SpO_2_ < 80% was found; episodes were also of significantly shorter duration while the infants received NAVA [15]. This may thus be a valuable addition to treating AOP via nasal respiratory support systems, but still requires further study.

Thus, although our data showed a smaller decrease in the rate of IH events than observed previously, they were in line with these previous studies in that we also found less IH and fewer bradycardias during s-NIPPV compared to cf-NCPAP.

## 5. Conclusions

Given the above limitations of our study, particularly its small sample size, the comparatively small differences observed between the three different modes of nasal respiratory support investigated should not prevent clinicians from preferring s-NIPPV or vf-NCPAP over cf-NCPAP when trying to prevent or treat IH episodes in very preterm infants.

## Figures and Tables

**Figure 1 children-07-00168-f001:**
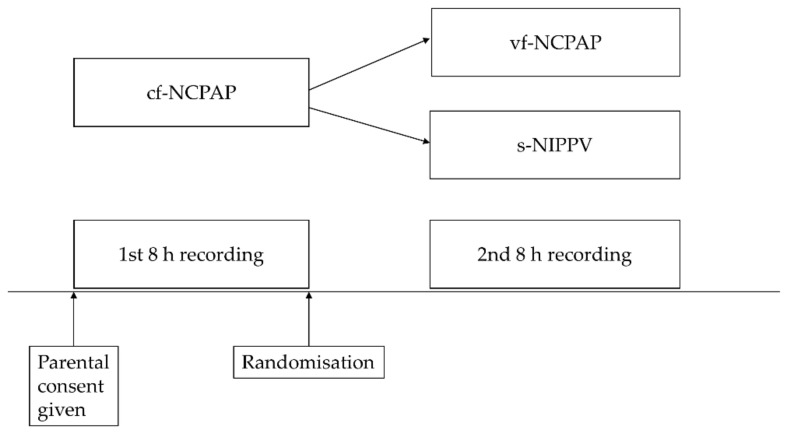
Study design; NCPAP: nasal continuous positive airway pressure; cf: continuous flow; vf: variable flow; s-NIPPV: synchronized nasal intermittent positive pressure ventilation.

**Figure 2 children-07-00168-f002:**
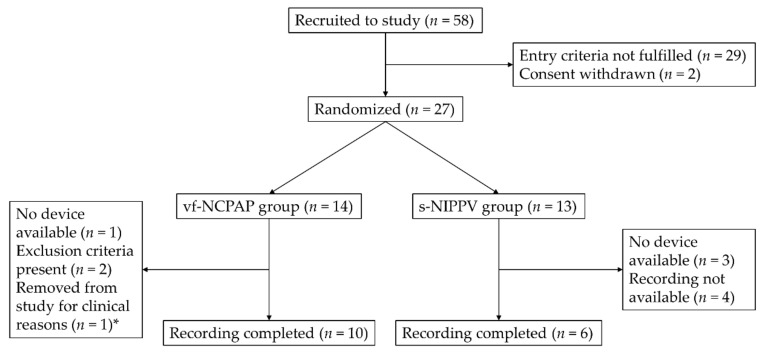
Patient flow; vf-NCPAP: variable flow nasal continuous positive airway pressure; s-NIPPV: synchronized nasal intermittent positive pressure ventilation; * the attending physician decided that changing the ventilator was not advisable.

**Table 1 children-07-00168-t001:** Patient demographics expressed as median (minimum–maximum).

	Total Group (*n* = 16)	vf-NCPAP (*n* = 10)	s-NIPPV (*n* = 6)
Male sex (n)	4	3	1
GA at birth (w)	27.5 (24.9–29.7)	27.0 (24.9–29.7)	28.4 (27.0–29.1)
Birth weight (g)	755 (590–1050)	718 (590–987)	940 (620–1050)
PMA at study (w)	30.5 (27.3–31.9)	30.2 (27.3–31.9)	30.5 (29.4–31.1)

GA: gestational age; PMA: postmenstrual age.

**Table 2 children-07-00168-t002:** Primary and secondary endpoints expressed as median (minimum–maximum).

	Before–After Study 1 (*n* = 10)		Before–After Study 2 (*n* = 6)	
	cf-NCPAP	vf-NCPAP	cf-NCPAP	s-NIPPV
IH rate (*n*/h)	1.38 (0–2.5)	1.06 (0.13–2.9)	0.88 (0–2.1)	0.69 (0–1.6)
Bradycardias (*n*/h)	0.56 (0–1.3)	0.25 (0–1.1)	0.81 (0.5–0.9)	0.44 (0.1–0.5)
Mean FiO_2_	0.23 (0.21–0.28)	0.22 (0.21–0.29)	0.21 (0.21–0.25)	0.21 (0.21–0.27)
Mean tcpCO_2_ (mmHg)	57.2 (50–77)	53.8 (46–72)	61.3 (48–64)	54.3 (50–64)

IH: intermittent hypoxia; FiO_2_: fraction of inspired oxygen; tcpCO_2_: transcutaneous partial pressure of carbon dioxide.

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
