# Peer review of "Randomized Longitudinal Study Comparing Three Nasal Respiratory Support Modes to Prevent Intermittent Hypoxia in Very Preterm Infants"

_children, 2020, doi:10.3390/children7100168_

Round 1

Reviewer 1 Report

This study compared 3 diff. nasal resp. support modes, which are used to prevent intermittent hypoxia in very preterm infants. Unfortunately the number of included infants  is very small, so it was not possible to compare the 2 intervention groups (vf-NCPAP vs. s-NIPPV), as the Group size was 10 and 6, respectively.

Introduction:

good and easy to read.

Methods:

Inclusion and exclusion criteria are clear.

Page 2 line 52, three points were given, if the Baby had to be turned over in preparation for bagging. Does this mean all infants sleep in a prone position for the whole time?

Page 2 line 59: Is it right that standard NCPAP was always cf-NCPAP and all infants started with cf-NCPAP? If so, could not this  be added to the inclusion criteria? Is it right that all infants had an eight hour recording on the cf-NCPAP before they were switched to the randomized mode, unless they had an apnea score of >= 10 within the 8 hour period? If so they changed the mode before 8 hours? For me, the randomization protocol is not clear yet in your manuscript, as it looks like you changed the mode at the latest of 8 hours, regardless of the apnea score. Maybe you can add more clarity.

Page 2 line 69: Did all infants had the same amout of caffeine base (5mg/kg/d) and no one got a loading dose? Did anyone get Doxapram?

Page 3 line 77: you defined IH as SpO2 <80% for more than 1 sec. and bradycardia <80bpm for more than 1 hb. For me this is a very wide definition, have you thought about categorizing the events regarding their duration and then comparing the groups? Is there any difference in the Duration of Ich on/or bradycardia using the different devices.

Results:

You found no clinically relevant changes in the rate of IH, however the included infants already had a very low rate of IH during the standard NCPAP therapy, from min. 0 to max. 2.9 events per hour, with a median of about 1. You have included very preterm infants, but the recordings of your study started at around 2 weeks of PMA. Maybe the included infants were too stable to find changes, especially with the small sample size? Also FiO2 was very low in the included infants, which again indicates a certain stability of the infants. How many of the infants or how often had the infant  be turned over and be ventilated?

Discussion:

Page 4 line 128: again qualification for randomization. Please see above.

Reviewer 2 Report

Main concerns:

  1. No mention of failure rate- how many infants "failed" vf-CPAP and s-NIPPV required invasive support? I assume it was zero but would be nice to know
  2.  Inclusion included infants of more mature gestation than would be expected. Apnea of premature improves in most infants by 34 weeks.
  3.  How was your apnea score validated? Previous publication? Seems > 10 score was out of proportion to otehr apnea/bradycardia events.
  4.  Dose of caffeine seems low @ 5 mg/kg

Round 2

Reviewer 1 Report

Dear Authors,

thank you for your clear response and the implication of some comments. Consequently the manuscript has improved.

All questions have been answered adequately.

Author Response

Author’s response to reviews

Title: Randomized longitudinal study comparing 3 nasal respiratory support modes to prevent intermittent hypoxia in very preterm infants

Authors: Maximilian Gross, Anette Poets, Renate Steinfeldt, Michael S. Urschitz, Katrin Böckmann, Bianca Haase and Christian F. Poets

Version: 2

Date: 17 September 2020 (version 1); 28 September 2020 (version 2)

Dear Professors Acra and Lee,

Dear Reviewer,

Thank you again very much for considering our revised manuscript for publication in a special issue on Neonatal Health Care of your journal.

We would like to thank you for your constructive criticism and are glad that this helped to improve our manuscript.

Sincerely,

Christian F. Poets on behalf of the co-authors

Department of Neonatology

University Children’s Hospital Tübingen

Calwerstr. 7

72076 Tübingen - Germany

Tel: +49 7071 / 29 - 80895

Fax: +49 7071 / 29 - 3969

E-mail: christian-f.poets@med.uni-tuebingen.de

Reviewer 2 Report

I again feel there is inherit limitations to this study. Authors continue to discuss reasons why the study was halted early, but it may be good to discuss limitation of s-NIPPV device. Author describe improved s-NIPPV with NI-NAVA approach. Another limitation not discussed is mean GA @ enrollment is GA 30 weeks. This GA may be to late to detect a difference in IH episodes.

Lastly, none of the babies in any group required or treated with PEEP > 5 indicating another limitation of the study was lack of obstructive apnea treatment.
